# The Effect of Parental Beliefs on Post-Traumatic Symptoms of the Parent and Child after the Child’s Surgery

**DOI:** 10.3390/children9081265

**Published:** 2022-08-22

**Authors:** Amichai Ben-Ari, Yael L. E. Ankri, Roy Aloni, Orly Buniak-Rojas

**Affiliations:** 1Department of Behavioral Sciences, Ariel University, Ariel 4070000, Israel; 2Herman Dana Division of Child and Adolescent Psychiatry, Hadassah-Hebrew University Medical Center, P.O. Box 12000, Jerusalem 9112001, Israel

**Keywords:** pediatric medical traumatic stress (PMTS), pediatric surgery, parental beliefs, risk factors

## Abstract

In recent years, many studies have attempted to find the main predictors of the development of post-traumatic symptoms in children following medical procedures. Recent studies found a link between parental beliefs and children’s post-traumatic symptoms in various medical contexts such as life-threatening illness, pain, and hospitalization. This study aims to examine the relationship between parental beleifs and post-traumatic symptoms in children and parents after surgical interventions of the children. The study was conducted among 149 children who underwent surgery and their parents. The children and parents were examined at 2 time points- during hospitalization, and 4 months after the hospitalization. Questionnaires were administered measuring parental beleifs pertaining to parental distress, and post-traumatic symptoms among children. results show a correlation between the factors. In addition, it was found that the parents’ distress is a mediating relationship between the parents’ perceptions and the child’s level of distress. It has been found that there is a link between some of the parental beleifs and parental stress symptoms and post-traumatic symptoms in the children. Parental beliefs that were found to influence these variables were related to parental beliefs regarding children’s suffering and pain during surgery. In addition, children of parents with higher levels of religious and spiritual beliefs were found to have fewer post-traumatic symptoms. This study sheds light on parental beliefs that may have the power to influence parental stress levels and children’s post-traumatic symptoms after surgery.

## 1. Introduction

Diseases and injuries of children can lead to medical treatment including surgical interventions. These interventions are accompanied by pain, uncertainty, helplessness, and even a threat to the child’s life [1]. In some cases, these medical events may meet the definition of a traumatic event, and children may develop post-traumatic symptoms. These symptoms have a significant effect on mental and physical rehabilitation and the functioning of children in all areas of life [2]. Although 25–30% of children experience post-traumatic stress disorder after a significant medical event, only about half meet the full criteria for PTSD [3]. As a result, post-traumatic stress disorder that develops as a result of medical events has been recognized as an independent and separate phenomenon from post-traumatic stress disorder (PTSD) and has been defined by the National Network for Post-Traumatic Stress Disorder in Children as pediatric medical traumatic stress (PMTS) [4]. PMTS includes a wide range of psychological and physiological responses of children and their families to pain, injury, serious illness, medical intervention, and invasive medical care. Younger children, who have a severe illness, and who have been exposed to many invasive procedures, appear to be at increased risk of developing adverse psychological consequences [2,5]. However, there are few studies that have addressed the development of post-traumatic symptoms specifically in children who have undergone surgical intervention [4].

Parents of children with chronic illness face various consequences of the illness and simultaneously most face intensive care for their child following the illness. Parents deal with increased emotional distress after the detection of the disease as well as with feelings of uncertainty and perhaps even the fear for their child’s life. As a result, these parents experience high levels of stress [6,7,8,9] and are at increased risk of developing anxiety disorders, depression, PTSD, and adjustment disorders [7,10]. There are many factors that can affect parents’ ability to adapt to the new situation after their child’s illness, or instead the risk of developing anxiety or a stress disorder. Previous studies have found that one of the reasons for adaptability lies in the parent’s beliefs [8,11].

In recent years, attention to the impact of cognitive beliefs and perceptions on the psychological and rehabilitative consequences after surgery and hospitalization has increased. Studies have addressed the effects of the interaction between disease perception and the patient’s core beliefs on psychological symptoms after surgery [12,13,14]. Core beliefs are defined as a general set of beliefs and assumptions about the world, the self, and relationships to others. After a traumatic event, an individual’s core beliefs are shattered and become shattered, which can lead to the development of distress [15]. However, challenging core beliefs can allow a person to re-examine their beliefs and consider the positive consequences of a traumatic experience thus leading to post-traumatic growth (PTG). For example, in a qualitative study conducted among cancer survivors, they reported that the disease was traumatic and confronted them with the real possibility of death. Yet survivors were still able to process the experience and challenge their beleifs so that their new perceptions of supportive social relationships or the sense of meaning in their lives led to post-traumatic growth [16]. It seems that shattering core beliefs can lead to both PTSD and PTG. When processing the experience and core beliefs are actively done by the individual it is possible that the individual will be able to grow as a result. In contrast, when the individual attempts avoiding thoughts about the traumatic event the distress levels can rise and be experienced as overpowering manifesting in PTSD symptoms [17].

Parents of children with cancer reported that the fact that the disease experience was understandable and manageable helped them cope during their child’s illness. It has also been found that religious beliefs contribute to the parent’s adaptation abilities by finding meaning during the illness, thus dealing with the lack of control experienced during the illness [8]. As opposed to adaptive beliefs that can be helpful, negative beliefs can be harmful. For example, parents who believed that their child would die during treatment or that he or she might still die of cancer (years after the end of treatment) reported increased post-traumatic stress symptoms. In addition, a catastrophic perception of pain has been found to be one of the most powerful factors in predicting high levels of experiencing pain in children and as a significant stressor [18]. The child’s expressions of pain can also, potentially, provoke considerable parental distress, which in turn can affect the child’s level of pain and distress [18,19,20]. Studies have shown that distress maternal distress in response to medical procedures clearly predicts the child’s distress [19].

Many studies have attempted to find the main predictor of the development of PTSD symptoms in children who have undergone medical procedures. For example, it was found that a high level of parental stress during hospitalization predicts a child’s emotional distress 3–6 months after discharge [2,5,6]. In addition, a correlation has been found between the parent catastrophic beliefs and post-traumatic symptoms in children suffering from chronic pain [21]. Thus, it seems that understanding the distress of the parents during hospitalization of their child will be beneficial not only for treating the immediate distress of the parent but also for promoting optimal emotional adjustment for the child and parent after the hospitalization.

In conclusion, the studies reviewed show that surgical intervention and hospitalization of a child is a stressful situation with potentially negative psychological consequences for the child [2,10,22]. Child hospitalization also affects the parent, and it has been found that a significant proportion of parents develop symptoms of distress after their child is hospitalized [6,8,9,10,23]. It seems that parent’s beliefs about sick children may affect the development of their children’s post-traumatic symptoms [5,6,21]. However, the literature lacks information regarding the correlations between the parent’s beliefs and the post-traumatic symptoms of the parent to the children’s post-traumatic symptoms after surgical intervention. The aim of the present study is to examine these correlations, shed light on this important issue and increase awareness regarding the influence of parents on the child undergoing surgery and thus reduce the rate of distress in this population.

## 2. Methods

### 2.1. Participants

One hundred and forty-nine parent-child dyads from a pediatric surgical ward were included in the study: 94 boys (63.1%), 55 girls (36.9%), and one parent per child participant, or 60 (40.3%) fathers and 89 (59.7%) mothers. Children’s age ranged between 1–6 years (M = 2.89, SD = 1.51). Of the children, 102 had been hospitalized for elective surgery (68.4%), and the rest for emergency interventions (n = 47, 31.5%). Hospitalization ranged from 1–37 days (M = 4.71, SD = 5.70). Demographic and hospitalization data (gender, cause of admission, the number of days of hospitalization) of the sample group were similar to those of the overall ward population between ages 1 to 6 (N = 6231) during the year. Consequently, it can be assumed that the sample of the study represents the general population of those hospitalized in the ward for that year (see Table 1). All the parents of the children participating in the study signed an informed consent form. The research was carried out by two psychology graduate students.

The sampling of subjects in the study was done using a “convenience sampling” method. Accordingly, all subjects who agreed to participate in the study were sampled from the population of parents whose children were hospitalized in a pediatric surgical ward, with the exception of parents of children who underwent neurosurgical intervention as well as non-Hebrew speakers, since the questionnaires were delivered in the Hebrew language. About 20 subjects dropped out during the study. All subjects participated in the study voluntarily.

The study was conducted after the approval of the Helsinki Committee and was carried out at the Hadassah medical center in Jerusalem [0437-14-HMO].

### 2.2. Measures

*Demographic questionnaire*: Demographic variables included age and gender of the child, and participating parent, level of religiosity, and years of education.

*The Family Illness Beliefs Inventory**(FIBI)* [8]: This questionnaire was developed by Kazak in 2004 and examines the family beliefs of parents of sick children. The questionnaire includes 4 items rated on a 1–4 Likert scale. Parents were asked how much they trust physicians (“the doctors will know what to do”), how much they believe the child will be in great pain (“my child will be in a lot of pain”), how much they trust their ability to make good therapeutic decisions (“we can make good therapeutic decisions”) and how optimistic they are about the child’s possibility to recover(“we’re going to overcome it”). Three of the questions deal with trust (in the system, in the parent, and in the child) and one question deals with the pain of the child. The questionnaire was translated into Hebrew by the researchers using the double translation method [24].

*Posttraumatic Stress Diagnostic Scale (PDS)* [25]: This questionnaire was administered in order to assess the level of distress of the parents (it is a self-report questionnaire). The PDS questionnaire includes 48 items (on a scale of 0–3) and is widely used in the research field and the clinical field. and has reported internal consistency of *a* = 0.92. We used the Hebrew translated and validated version [26]. In the current study, internal reliability yielded *a* = 0.97.

*UCLA PTSD Reaction Index for DSM-5 Parent/Caregiver Version for Children Age 6 Years and Younger* [27]: This is a questionnaire developed in 1985 in order to assess post-traumatic stress among children and adults. It is a PTSD assessment tool that has a high level of reliability and validity (Cronbach’s α = 0.88–0.91), without a significant level of variation between groups of different ethnic origins or religious backgrounds. The results of this measure are consistent with the results of other PTSD questionnaires such as the PTSD Checklist, PTSD Symptom Scale, and Harvard Trauma Questionnaire. The internal reliability scale (Cronbach’s α) in the present study was α = 0.95.

*Young Child PTSD Checklist (YCPC)* [28]: It is a self-report tool for assessing post-traumatic stress disorder in preschool children (ages 0–6). The questionnaire is filled out by the child’s parents and includes 42 items (13 items about the traumatic event, 23 items about the symptoms the child suffers from, and six items about the functional impairment the child suffers from).This questionnaire was found to have high reliability (Vasileva & Petermann, 2017) (arousal: Cronbach’s α = 0.77, avoidance: Cronbach’s α = 0.79, reliving: Cronbach’s α = 0.81), indicating its effectiveness in diagnosing PTSD in preschool children. In our study, good internal reliability for symptom scales (Cronbach’s α) were found, arousal: α = 0.92, avoidance: α = 0.93, reliving: α = 0.88, total score: α = 0.97. In addition, good internal traceability for the functioning scale was also found: α = 0.96.

### 2.3. Procedure

The research procedure consisted of two stages.

(1)From April to July 2018, out of 204 children consecutively hospitalized, 184 parents agreed to participate in the study (0437-14-HMO). Six parents refused to attend due to emotional reasons and 28 parents due to lack of time. An explanation of the study was provided to parents, parents signed a consent form and completed the Family Illness Beliefs Inventory and the demographic questionnaire.(2)In the second phase, four months after hospitalization, 149 parents completed the UCLA [27] and YCPC [28] questionnaires (assessing medical trauma and PTMS), and the PDS [25] questionnaire (assessing the parent’s distress level), with the help of researchers during a face-to-face interview.

As is customary in using structured and valid questionnaires, the questionnaires were administered by psychology master’s degree students who were specially trained for this purpose. Prior to administering the questionnaires, a pilot phase was conducted in which the questionaries were administered under guidance of an expert in order to make sure that the questionnaires were delivered properly.

### 2.4. Data Analysis

Data were analyzed using SPSS v.27 (IBM, Chicago, IL, USA). Descriptive statistics have been produced using frequencies for categorical variables and means with standard deviations for continuous variables. First, we performed a factor analysis for the two PTSD questionnaires in order to examine the possibility of gathering variables into one variable. Second, we assessed associations between variables by using Pearson correlations. Finally, to assess the mediation model, we performed the Structural Equation Modeling (SEM). The following indices were used to evaluate the model: chi-squared, which is acceptable when the value is not significant; the goodness of fit index (GFI), the comparative fit index (CFI), and Tucker-Lewis index (TLI), (adequate values—above 0.90, excellent fit—above 0.95); and the root mean square error of approximation (RMSEA) (adequate values—less than 0.08, excellent fit—less than 0.06) [29]. Level of significance (*p*-value) was 5%.

## 3. Results

First, it should be noted that the distribution of the obtained data follows a normal distribution by Skewness and Kurtosys test [30]. Second, in order to rate the quality of the correlation we will use the following index as is customary in studies in the field of behavioral sciences: A correlation between 0 to ±0.1 was considered as no correlation. A correlation between 0.1 to ±0.29 will be considered a weak correlation. A correlation between ±0.3 to ±0.49 will be considered a moderate correlation. A correlation between ±0.5 to ±1 will be considered a strong correlation [29].

As displayed in Table 2 correlations between background variables and variables predicted in the data set (parental distress and PTSD level), in order to examine the need to monitor these variables when testing the hypotheses. The correlations with quantitative or dichotomous background variables were calculated using the Pearson correlation, and the correlations with the order variables from the order scale were examined using the Spearman correlation. As can be seen, positive and significant associations were found between duration of hospitalization and parental distress and post-traumatic symptoms, so that the longer the hospitalization, the higher the parental distress, and the stronger the symptoms. In addition, a significant negative relationship was found between the level of religiosity and the level of distress of the parents, so that the higher the level of religiosity, the lower the distress of the parents. Finally, a significant positive correlation was found between the difficulty of the surgery and post-traumatic symptoms, so that the more complex the surgery, the higher the symptoms.

Following these results, in follow-up analysis the length of hospitalization and level of religiosity were included as monitored variables when predicting parental distress, and the length of hospitalization and the complexity of the surgery were included when predicting PTSD.

As a first step in testing the research hypotheses, Pearson correlations between the variables were calculated (Table 3). Consistent with the research hypothesis, it was found that there is a positive correlation between the parent’s perception that “the child will be in great pain” and the level of parental distress. However, no significant correlation was found between the belief that “the doctors will know what to do”, the belief that the parents “can make good therapeutic decisions” and the belief that they are “going to overcome it” to the level of parental distress. In addition, it was found that the belief that “the doctors will know what to do” is correlated with a lower level of post-traumatic symptoms (according to the YCPC questionnaire), and that the perception that “the child will be in great pain” is positively correlated with post-traumatic symptoms. The belief that they are “going to overcome it” was not found to be correlated with the level of post-traumatic symptoms.

According to the mediation hypothesis, parental stress is a mediating factor in the relationship between parental beleifs and post-traumatic symptoms. In order to test this hypothesis, the Structural Equation model was used, using the AMOS program. In accordance with the results presented in the preliminary tests, the duration of hospitalization, the complexity of the surgery and the level of religiosity were monitored. Table 4 shows the quality indices of the match between the theoretical model and the empirical model. The calculated Chi-Squared value was found to be significant, indicating a good match between the study data and the approximate model. Similarly, the GFI, TLI, CFI indices were found to be particularly high, and also indicate a high fit. The RMSAE index is found to be sufficiently low. Details of the model paths are shown in Figure 1.

As can be seen from Figure 1, the path between the perception that the child will be in great pain and the parental distress is found to be positive and significant. The rest of the beleifs were not found to clearly predict the level of parental stress. In addition, the pathways from the level of parental distress to PTSD scores were also found to be positive and significant. Accordingly, the mediation path between the perception that “my child will be in great pain” to score a UCLA questionnaire was found to be positive and significant, Effect = 0.03, 95% CI = 0.002, 0.07, as well as the mediation path between “my child will be in great pain” and the YCPC score, Effect = 0.03, 95% CI = 0.004, 0.08 These results support that the mediation hypothesis was partially supported. Support has been found that the more the parents believe that their child will be in pain, the greater their parental distress, and the greater the parental distress, the higher the post-traumatic symptoms.

## 4. Discussion

This study is based on a sample of 149 parents of children after surgery. The study examined the relationship and impact of parental beliefs on the parent’s and child’s post-traumatic symptoms after surgery. The aim of the study was to examine which parental beliefs affect the child’s post-traumatic symptoms, and whether there are debating or moderating factors that may affect the child’s post-trauma. For the purpose of conducting the study, 4 parentals beliefs were examined, representing 4 different factors from Kazak’s study [8] on parental beliefs and their effect on parental distress. A positive correlation was found between the parent’s parental beliefs regarding the suffering and pain associated with treatment and the parents’ distress symptoms. This finding links to a previous study, which examined negative beliefs and found that parents who believed their child would die or suffer during treatment were found to have increased post-traumatic stress symptoms [31]. However, it has not been found that parental beliefs are related to the ability to trust the physician’s ability and the belief in recovery are related to symptoms of distress and parental distress. It is possible that the difference between these parental beliefs and the perception of pain and suffering in the child lies in the ability to challenge the parent’s beliefs [16]. While the parent’s perception of the child’s pain and suffering during surgery appears to cause the parent to experience distress symptoms such as avoidance and intrusion, the same effect was not found with the beliefs pertaining to the belief in the medical staff’s ability, and belief in recovery. This finding is important for 2 reasons: The first is the ability to make a distinction between parental perceptions and beliefs that are relevant to medical procedures, such as surgery and might affect them. This point is important since not every parental perception affects the parent and child, and it is important to identify those that do. The second reason is recognizing and emphasizing parental beliefs related to the child’s suffering. Parental beliefs regarding a child’s pain appear to be broad-based, and may be a risk factor for parental distress, resulting in the child developing post-traumatic symptoms.

Another interesting finding in the study is related to the level of religiosity of the parents. A negative correlation was found between the level of religiosity and the level of parental distress so that the more religious the parent was the lower his level of distress was. This finding is consistent with previous studies, showing that religious beliefs and spiritual beleifs may contribute to parent’s adjustment during the period of their child’s illness, thus influencing the level of parental stress [8]. This finding is also consistent with previous studies that have found that a high level of spirituality reduces invasive thoughts, reduces emotional distress, and contributes to post-traumatic growth after a serious illness [32,33]. Given findings from previous studies that the lower the level of parental stress, the more PTSD symptoms can be reduced in the child, it can be concluded that spirituality moderates the relationship between the two through this mechanism.

According to Kazak [8], there are 4 factors that are influenced by parental beliefs and may affect a parent’s stress symptoms: suffering related to treatment, trust in the physician’s ability, sense of ability, and belief in recovery. A significant positive correlation was found between the parent’s beliefs regarding suffering of the child in the treatment and the post—traumatic symptoms of the child. These results demonstrate the correlation between parent’s beliefs and parental distress. In addition, this finding is related to the mediation model detailed below.

Regarding the factor 1 associated with suffering as a result of treatment, support was found that the stronger the parents’ belief that their child will be in pain as a result of the surgery the higher the parental distress, and that higher distress increases the child’s post-traumatic symptoms. This finding is consistent with previous studies, which found that among all the predictors, parental stress symptoms during hospitalization predicted adverse emotional changes in hospitalized children between three and six months after discharge [5]. Further studies have found that parental traumatic stress has been shown to predict post-traumatic psychological distress in children [2,6]. The novelty of this finding, however, is that the cause of post-traumatic symptoms is the parent’s perception that the child will be in pain during surgery and is mediated by the parental distress associated with this perception. Previous studies have shown that parenting perception regarding pain related to parental distress [19], this study shows that not only was the parental perception regarding pain related to parental distress, but that parental distress affects the child’s post-traumatic symptoms.

However, this study found no mediating relationship between parental beliefs regarding medical staff abilities, beliefs of parental ability, and belief in recovery. Kazak [8] examined four factors related to parental stress symptoms, but it appears that although the four factors are related to parental distress, only one factor is also related to post-traumatic symptoms in the child as a result of parental stress. This may be because factor 1 is the only factor related to the child himself and not to the environmental or social factors related to the surgery. It is possible that the perception that directly concerns the child’s physical condition is the most influential on the level of parental distress and the child’s post-traumatic symptoms, then that concern doctors or the environment.

Finally, Previous studies have examined the relationship between the parent’s catastrophic thinking regarding the child’s pain and found that the higher the parent’s tendency towards catastrophic thinking, the greater the child’s distress regarding pain [18,19,20]. The novelty and importance of the findings in this study is in the findings that the impact of parental perception on the child’s pain after surgery. This study demonstrates that this perception is directly related to the level of parental distress, and as a result has been found to be associated with post-traumatic symptoms in the child. Previous studies show that 16–28% of children will develop post-trauma symptoms with a significant effect on their physical recovery after a medical event. Post-traumatic symptoms have a significant impact on the functioning of these children in all areas of life [2]. Proper identification of the factors that may increase post-traumatic symptoms in the child after surgery is necessary, especially when correct identification will lead to the possibility of performing proper interventions, thus assisting both parent and child prior to surgery.

## 5. Conclusions

The main conclusions from this study are related to the potential link between certain parental beleifs and post-traumatic symptoms in the child, after surgery. First, this study reinforces previous findings in the literature that claim that parental distress affects post-traumatic symptoms in children [2,6]. Parental distress was also found in this study to be positively and significantly related to post-traumatic symptoms children.

Second, it seems that not all parental beliefs have the same affects. Parental beleifs related to the child’s pain and suffering during surgery were significantly associated with post-traumatic symptoms in parents and children. In contrast, parental beliefs in the physician’s ability their own sense of ability, and belief in recovery were not found to be significant or to affect parental distress and the child’s post-traumatic symptoms.

Based on the results of the study, three courses of action can be recommended to reduce parental distress associated with the child’s pain, thereby reducing the child’s risk of development of post-traumatic symptoms. The first recommendation is to use sedation or anesthesia to reduce the child’s pain during surgical procedures. This procedure should include an explanation to the parents emphasizing that the surgical procedure will not hurt the child at all as a result of the anesthesia. The second recommendation is to have a therapeutic meeting between the parents and a professional before the surgery, so that the parents can better deal with the fear of pain, thus ultimately reducing post-trauma symptoms in their child [4]. The third recommendation is in accordance with the finding that spirituality assists in reduction of parental stress symptoms. It is therefore recommended that prior to surgery parents may be given ample time to perform spiritual based actions, such as meditation or prayer.

The present study has two limitations. The first is related to the analysis of the subjects—when the subjects were sampled, the sample included both children who have undergone emergency surgery as well as elective surgery. It might be necessary to separate the two groups since there may be significant differences in the level of initial stress between emergency and elective surgeries. The second limitation is related to the homogeneity of the participants. The subjects were sampled among Hebrew speakers only, and in only one Jerusalem hospital. It may be possible that the sample is not a completely representative sample and may have certain biases.

## Figures and Tables

**Figure 1 children-09-01265-f001:**
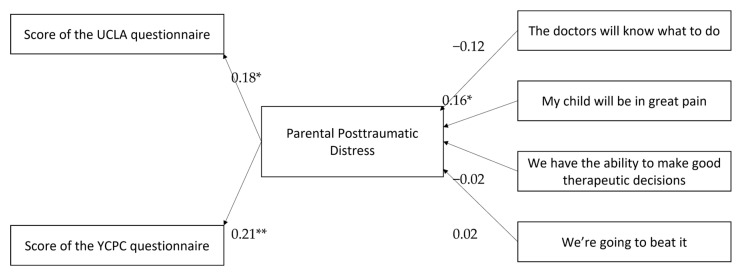
Summary of the mediation model the relationship between parental beleifs and post-traumatic symptoms by parental distress. (*) = Statistically significant relationship with confidence level of >0.5. (**) Statistically significant relationship with confidence level of >0.01.

**Table 1 children-09-01265-t001:** Demographic and clinical background characteristics of the sample (N = 149).

	*M*	*SD*	*Min.*	*Max.*	*n*	*%*
**Age**	2.86	1.51	1	6		
**Years of father’s education**	14.90	3.46	12	32		
**Years of mother’s education**	14.59	2.81	12	35		
**Duration of hospitalization**	2.60	2.03	1	12		
**Para-morbid parental distress**	1.28	0.38	1	3		
**Level of religiosity**						
Secular					21	14.1
Traditional					20	13.4
Religious					24	16.1
Ultra-Orthodox					84	56.4
**Economic difficulties**						
None					110	77.5
Certain economic problems					27	19
Many economic problems					2	1.4
Difficulty in basic needs					3	2.1
**The degree of difficulty of the surgery**						
Low					89	63.6
Moderate					46	32.9
Hard					5	3.6
**Exposure to previous trauma**						
No					137	91.9
Yes					12	8.1

**Table 2 children-09-01265-t002:** Correlations between background variables to PDS and PTSD scores (N = 149).

	Parental Distress	UCLA Questioner	YCPC Questioner
**Pearson Correlation**			
Age	−0.10	−0.03	−0.05
Years of father’s education	−0.10	−0.08	−0.04
Years of mother’s education	−0.04	−0.10	−0.10
Duration of hospitalization	0.26 *	0.29 **	0.47 **
Para-morbid parental distress	0.11	0.08	0.10
Exposure to previous trauma (no, yes)	−0.07	0.14	0.09

* *p* < 0.05. ** *p* < 0.001. Note: Relationships between quantitative and dichotomous variables were examined using Pearson correlations, relationships between variables from an ordinal scale were examined using Spearman correlations.

**Table 3 children-09-01265-t003:** Pearson correlations between study variables (N = 149).

	1	2	3	4	5	6	7
1. The doctors will know what to do	-						
2. My child will be in a lot of pain	−0.12	-					
3. We can make good therapeutic decisions	0.33 **	0.11	-				
4. We’re going to overcome it	0.22 **	−0.05	0.30 **	-			
5. Parental distress	−0.08	0.23 **	−0.01	0.05	-		
6. UCLA score	−0.13	0.35 **	0.09	−0.10	0.21 **	-	
7. YCPC score	−0.14 *	0.33 **	0.12	−0.10	0.26 **	0.90 **	-

* *p* < 0.05. ** *p* < 0.01.

**Table 4 children-09-01265-t004:** Matching quality indices of the approximate model (N = 149).

Model	$\chi^{2}$	*df*	*p*	*CFI*	*TLI*	*GFI*	*RMSEA*
**I**	1.65	16	0.048	0.980	0.904	0.975	0.066

Note. CFI = comparative fit index; TLI = Tucker–Lewis index; GFI = Goodness of Fit index. RMSEA = root-mean-square error of approximation.

## Data Availability

For the complete database please contact by email: baamichai@gmail.com.

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
