# Peer review of "The Effect of Parental Beliefs on Post-Traumatic Symptoms of the Parent and Child after the Child’s Surgery"

_children, 2022, doi:10.3390/children9081265_

Round 1

Reviewer 1 Report

The article 'The effect of parental beliefs on post-traumatic symptoms of  the parent and child after the child's surgery' by Ben-Ari et al is significant sudy to mitigate parental stress levels and post-traumatic symptoms in the child after surgery. The methodology and results presented are convincing and clear. 

The authors should elaborate on how parents perception of religiosity moderate child's post-trauma, and suggest any measures for instance meditation to overcome the stress

I recommend the arcticle to be published after suggested revision

Author Response

We thank the reviewer for the comment. Regarding the spiritual views of the parents, this topic was addressed on page 11 (in the part highlighted in green). However, in light of the comment we have added an additional reference (highlighted in light blue).

In addition, we thank the reviewer for the suggestion to add a recommendation regarding meditation and we have added it in the recommendations section on page 14 (highlighted in light blue).

Reviewer 2 Report

I was pleased to read the article entitled "The effect of parental beliefs on post-traumatic symptoms of the parent and child after the child's surgery".

The abstract must be written in accordance with the guidelines of the journal.

It is awkward at the end of the Introduction to start the sentence with In conclusion. (Line 103)

How was the sampling done?

Do you think it is justified to observe elective and emergency patients at the same time?

Cronbach's α = 0.95.0 ? (Line 153)

Data of informed consents are missing! Was the interview by the researcher always conducted by the same person?

Also, the authors did not provide information on Funding, Data Availability Statement, Conflicts of Interest, etc.

Did the distribution of the obtained data follow a normal distribution? If so, write it.

You have not written anywhere how you will grade the results of the correlations. For which ranges do you think there is no correlation, or that there is, and whether it is good or excellent.

It is necessary to write the ethical approval of the study.

Do you think the study has limitations? If yes, write them.

The conclusion section is written too extensively. Cut it short.

In the conclusion section, there are a number of sentences that should be part of the discussion section.

Author Response

  1. The abstract must be written in accordance with the guidelines of the journal.

We accept the comment and have changed the abstract according to the guidelines of the journal.

  1. It is awkward at the end of the Introduction to start the sentence with In conclusion. (Line 103) 

We accept the comment and delete the words "In conclusion".

  1. How was the sampling done?

Following the comment, we added the information regarding the sample appearing in the subject's paragraph on page 5 (highlighted in yellow).

  1. Do you think it is justified to observe elective and emergency patients at the same time?

This is a valid comment. Indeed, there is a problem in comparing the 2 types of surgeries, so we decided to mention this point in the paragraph that talks about the limitations of the study. However, it should be noted that in the test we conducted using a T-test for independent samples, no significant difference was found between the 2 types of analysis in the measure of parental distress and in the two questionnaires of the children's distress.

  1. Cronbach's α = 0.95.0 ? (Line 153)

Fixed.

  1. Data of informed consents are missing! Was the interview by the researcher always conducted by the same person?

We added a reference to these questions on page 5 (marked in green).

  1. Also, the authors did not provide information on Funding, Data Availability Statement, Conflicts of Interest, etc.

This information appears on the first page (highlighted in yellow). Regarding the availability of the data, if this is required we can transfer you the data files of the study.

  1. Did the distribution of the obtained data follow a normal distribution? If so, write it.

We added a reference to this on page 7 (highlighted in yellow).

  1. You have not written anywhere how you will grade the results of the correlations. For which ranges do you think there is no correlation, or that there is, and whether it is good or excellent.

Following this comment we added a reference on page 7 (highlighted in green).

  1. It is necessary to write the ethical approval of the study

Following this comment, we added a reference to it on page 5 (highlighted in light blue).

  1. Do you think the study has limitations? If yes, write them

Following this comment, we added a paragraph of the limitations of the study on page 14 (highlighted in green).

  1. The conclusion section is written too extensively. Cut it short. In the conclusion section, there are a number of sentences that should be part of the discussion section.

We thank the reviewer for this comment. The text has been modified in accordance with this recommendation.

We would like to thank the Reviewer for the important comments that have helped us to further clarify our study, and you, once again, for your consideration of the manuscript.

Round 2

Reviewer 2 Report

According to the legal regulations of your country, you must write exactly who signed the informed consent, the parents or the children themselves.

Do you think the results are justified if the interviews were not conducted by the same person? Please clarify.

Please write which test you used to check the distribution of the obtained data.

O means no correlation. That can't possibly be a medium correlation! If you found the mentioned ranges in relation to an article, it would be good to write a reference.

It is awkward to start a sentence with "Table 2 shows..."

Instead of the word "Findings" write "Results"

I still think that the conclusion is too broad and that there are a number of sentences that can be part of the discussion.

The limitations section should be written at the end of the discussion section. A subtitle is not necessary.

Do you have the option to show results separately for emergency and elective patients? This would still give better data and conclusions, which would make the study better.

I recommend the authors to once again thoroughly revise the manuscript and check the spelling.

Author Response

We thank the reviewer for the important comments that will allow us to improve the article. Below is the detailed reference to the comments:

  1. According to the legal regulations of your country, you must write exactly who signed the informed consent, the parents or the children themselves.

    We added a reference to this at the beginning of page 6 (all corrections and additions in this version are highlighted in pink)

    2. Do you think the results are justified if the interviews were not conducted by the same person? Please clarify.

    We added a reference to this at of page 7 (all corrections and additions in this version are highlighted in pink).
  2. Please write which test you used to check the distribution of the obtained data.

    We added a reference to this at of page 7 (all corrections and additions in this version are highlighted in pink).
  3. O means no correlation. That can't possibly be a medium correlation! If you found the mentioned ranges in relation to an article, it would be good to write a reference.

    This is a valid comment. Following this comment it was corrected (appears on page 8 and highlighted in pink).
  4.  It is awkward to start a sentence with "Table 2 shows..."

    We fixed it on page 8 (highlighted in pink.
  5. Instead of the word "Findings" write "Results"

    We have corrected this throughout the article (highlighted in pink).
  6. I still think that the conclusion is too broad and that there are a number of sentences that can be part of the discussion.

    We decided to accept your comment and moved another significant part of the conclusions chapter (page 13) to the first paragraph of the discussion chapter (page 12). All changes are highlighted in pink.
  7. The limitations section should be written at the end of the discussion section. A subtitle is not necessary.

    Fixed it as per comment
  8. Do you have the option to show results separately for emergency and elective patients? This would still give better data and conclusions, which would make the study better.

    Unfortunately, we are unable to conduct such an analysis at the moment (in the near future) due to an availability problem of the statistician. However, we accept the comment regarding the difficulty in comparing children who underwent emergency surgery to children who underwent elective surgery, so we included a reference to this in the paragraph that talks about the limitations of the study.
  9. I recommend the authors to once again thoroughly revise the manuscript and check the spelling

Following this comment, we made a major linguistic edit to the entire text.
